# Embrace the Complexity: Agnostic Evaluation of Children’s Neuropsychological Performances Reveals Hidden Neurodevelopment Patterns

**DOI:** 10.3390/children9060775

**Published:** 2022-05-25

**Authors:** Elisa Cainelli, Luca Vedovelli, Dario Gregori, Agnese Suppiej, Massimo Padalino, Paola Cogo, Patrizia Bisiacchi

**Affiliations:** 1Department of General Psychology, University of Padova, 35131 Padova, Italy; patrizia.bisiacchi@unipd.it; 2Unit of Biostatistics, Epidemiology, and Public Health, Department of Cardiac, Thoracic, Vascular and Public Health Sciences, University of Padova, 35131 Padova, Italy; luca.vedovelli@unipd.it (L.V.); dario.gregori@unipd.it (D.G.); 3Department of Medical Sciences, Pediatric Section, University of Ferrara, 44121 Ferrara, Italy; agnese.suppiej@unife.it; 4Pediatric and Congenital Cardiovascular Surgery Unit, Department of Cardiac, Thoracic, and Vascular Sciences, Padova University Hospital, 35128 Padova, Italy; massimo.padalino@unipd.it; 5Clinica Pediatrica, Department of Medicine, University Hospital Santa Maria della Misericordia, University of Udine, 33100 Udine, Italy; paola.cogo@uniud.it; 6Padova Neuroscience Center (PNC), University of Padova, 35129 Padova, Italy

**Keywords:** preterm, hypoxic-ischemic encephalopathy, congenital heart disease, cluster analysis, machine learning, cognitive

## Abstract

The most common adverse pre/perinatal events have a great impact on neurodevelopment, with avalanche effects on academic performance, occupational status, and quality of life. Although the injury process starts early, the effects may become evident much later, when life starts to pose more challenging demands. In the present work, we want to address the impact of early insults from an evolutionary perspective by performing unsupervised cluster analysis. We fed all available data, but not the group identification, into the algorithm for 114 children aged 5–10 years, with different adverse medical conditions: healthy (n = 30), premature (n = 28), neonatal hypoxic-ischemic encephalopathy (n = 28), and congenital heart disease (n = 28). We measured general intelligence and many neuropsychological domains (language, attention, memory, executive functions, and social skills). We found three emerging groups that identify children with multiple impairments (cluster 3), children with variable neuropsychological profiles but in the normal range (cluster 2), and children with adequate profiles and good performance in IQ and executive functions (cluster 1). Our analysis divided our patients by severity levels rather than by identifying specific neuropsychological phenotypes, suggesting different developmental trajectories that are characterized by good resilience to early stressful events with adequate development or by pervasive vulnerability to neurodevelopmental disorders.

## 1. Introduction

Over the last decade, a dramatic increase in the incidence of neurodevelopmental disorders in childhood has been reported [1]: for example, diagnoses of attention deficit hyperactivity disorder (ADHD) have tripled, and diagnoses of autism have increased 20 times [2]. Among developmental disorders, specific learning disorders (reading, writing, and mathematics deficits) are the most frequently diagnosed disorders in children. Epidemiological studies report prevalence rates of 4% to 9% for reading deficits and 3% to 7% for mathematics [3,4].

Given their widespread incidence, disorders affecting children represent a significant health burden at the social and economic levels. The main challenges are often encountered in academic and educational contexts. For a small but growing percentage of children, the school readiness pathways are compromised: some children experience difficulties with impulse control, attentional capacity, and hyperactivity. Similar symptoms may hinder their ability to acquire crucial skills, such as focusing on teachers, interacting with peers and authority figures, and developing emergent literacy, mathematics, and language.

Given the extent of this phenomenon, attention to neurodevelopment, learning opportunities, and challenges has increased rapidly in the last decades. Worldwide efforts have been made to implement special services and school programs with the goal of developing child-friendly schools that consider children’s strengths and difficulties and build on and enhance individual potential.

The causes of neurodevelopmental disorders are complex and multifactorial. Unfortunately, the understanding of the etiopathogenic mechanisms remains rudimentary. Most studies point to the importance of events that occur during gestation and the first months of life [5], a phase of enormous quantitative and qualitative changes in the nervous system. During this period, the entire course of neurodevelopment is programmed and the determinants for future proficiency are set. Unsurprisingly, a high incidence of neurodevelopmental disorders affects children who have experienced insults during gestation or in the early postnatal periods. Furthermore, neurodevelopmental sequelae have also been reported in children who suffered from only mild or moderate pathological conditions during this phase [6]. Alarmingly, while advances in pre/perinatal medicine have reduced the incidence of mortality and severe morbidity, late-emerging minor impairments have been increasingly reported [7,8].

Early insults may trigger pro-inflammatory processes followed by a cascade of worsening events, which interact with multiple factors characterizing later life, determining various and complex scenarios. Infection or inflammation can sensitize the brain to injuries and make them more severe. Cytokines such as IL-10 are inflammatory mediators that readily cross the blood–brain barrier. They have a dual role of being protective and harmful, and their balance is critical in brain injury, even if their increments are non-specific. Among cytokines, IL-1beta promotes brain damage through reactive oxygen species rise, and, with IL-6 and TNF-alpha, its level is a proxy of white matter injury that could lead to cerebral palsy when measured in the amniotic fluid of women with complicated preterm gestation. Many pathways are incriminated in the apoptosis of premyelinating oligodendrocytes or subplate neurons involved in perinatal brain development at the molecular level. Rising glutamate concentrations or free radical reactive species (both oxygen and hydrogen) in HIE and inflammatory cytokines from activated microglia and astrocytes have been extensively mentioned in both white and grey matter injuries [9,10]. If not recognized quickly, these mechanisms impair the capacity of the immature brain for optimal development. This vulnerability becomes evident when the environment becomes more demanding, and complex cognitive abilities, such as learning and social relationships, are required, usually at school age. Therefore, preschool years become a critical developmental period during which the building blocks for later success are laid and the social, behavioral, and neuropsychological skills necessary for academic success are acquired [9].

Very common early stressful conditions, such as prematurity, neonatal hypoxic-ischemic encephalopathy (HIE), and congenital heart disease (CHD), increase children’s risk of developing a wide range of neurodevelopmental disorders and subtle diseases, such as neuropsychological or learning impairments [11]. These problems often are not evident in the first years of life but emerge at older ages in the absence of overt brain damage or clinical complications. Given the lack of severe impairment, the lack of confidence of the medical professional with neurocognitive problems, and the difficulty in managing such long-distance emerging symptoms, these children were often considered “normal”, “healthy,” or, in general, without problems. Therefore, they have not been followed and monitored on a long-term neurodevelopmental basis.

Currently, neurodevelopmental disorders represent a significant burden on the health system, families, and schools. Given that pathologies such as prematurity affect 10% of all births [12], CHD affects nearly 1% of all newborns as the most common congenital defect [13]. The incidence of neonatal HIE ranges from 1 to 8 per 1000 live births in developed countries [14], and the lack of awareness about long-term sequelae and the uncertainty of rehabilitative options is alarming.

The medical literature includes several reports on the neuropsychological development of children suffering from pre/perinatal insults. Despite heterogeneous etiopathogenesis, the most common adverse conditions in early life (prematurity, HIE, and CHD) can determine a wide range of neuropsychological impairments and learning difficulties without a defined characterization between medical conditions or specific phenotypes. Most frequently, complex high-order abilities, such as attention and executive function, are the first affected [7,15,16,17], as shown in studies on pediatric neurological conditions [18,19,20,21].

Given the specificity of the works and the medical perspective, comparing these studies is not easy, and a comprehensive understanding of brain developmental mechanisms is lacking. In the present work, we want to change the point of view and address the impact of early insults on neurodevelopment from an evolutionary perspective. For this reason, we investigated the possible emergence of different neuropsychological functioning profiles by assessing a large group of children with different medical conditions: both healthy and prematurely born children with diagnoses of HIE and CHD. We used unsupervised cluster analysis, feeding the algorithm all the data but not providing the groups/time points identification. We then analyzed the emerging neuropsychological profiles (clusters) for differences in medical status.

We hypothesize that a similar investigation might provide interesting information on developmental trajectories in children with a history of pre/perinatal insults, given their high risk of developing neurodevelopmental disorders. We excluded children with severe impairments who might have to introduce biases in data interpretation. Furthermore, understanding the neurodevelopment of children who do not suffer from severe neonatal conditions or develop severe disabilities is challenging because the long-term prognosis is difficult to know at birth and subtle cognitive deficits may not be recognized.

## 2. Materials and Methods

### 2.1. Participants

In this study, we used data obtained from three larger ongoing projects that conduct long-term monitoring of newborns admitted to neonatal intensive care units due to preterm birth (“preterm” group) and neonatal HIE (“HIE” group) and of children who have undergone elective surgery for CHD (“CHD” group). The study was carried out according to The Code of Ethics of the World Medical Association (Declaration of Helsinki) for experiments involving humans. The hospital’s ethics committee approved each project (details are available for each project in the following publications: [6,15,22]). Parents provided written informed consent for their children to participate.

Children included in this study were sequentially recruited from January 2010 to June 2015. Inclusion and exclusion criteria for the recruitment of premature children, HIE, and CHD have been previously described [6,15,22] and are reported in Appendix A.

For this study, we selected children aged 5 to 10 years with family consent to participate in the study. Exclusion criteria included the presence of major impairments (neurosensory impairments, cerebral palsy, and epilepsy); subsequent diagnosis of congenital malformations, inborn metabolism errors, genetic syndromes, or other medical comorbidities; intelligence quotient (IQ) < 70; traumatic events or documented parental neglect; and invalidating parental pathologies that emerged during clinical follow-up.

The number of children that participated in this study is not the same as the participants of the original studies; this is partly because not all participants in the initial studies reached the age of 5 years; additionally, some participants refused to participate or became unavailable.

Children without prenatal/perinatal risk factors (“healthy” group) were recruited at birth from the hospital’s nursery and longitudinally followed by the same projects.

### 2.2. Neuropsychological Assessment

We measured general intelligence using the Wechsler Preschool and Primary Scale of Intelligence III (WPPSI-III, [23]) test or the Wechsler Intelligence Scale for Children IV (WISC-IV, [24]), standardized for the Italian population [25].

We used the naming test for language [26]; for attention, the visual and auditory attention tests of the NEPSY-II [27,28]; for memory, the design memory test of the NEPSY-II [27,28], which evaluates short-term visuospatial memory; for executive function, the coding test of the WISC-IV or WPPSI-III [23,24] and the semantic verbal fluency test, which evaluates the ability to access the lexicon through a categorical cue [26]; and for social skills, the theory of mind A and B and affect recognition tests of the NEPSY-II [27,28].

### 2.3. Statistical Analysis

The raw neuropsychological results were converted into standardized scores using appropriate published normative data, which accounted for the different age ranges. Data were expressed as median (interquartile range) and frequency (%). The nonparametric Kruskal–Wallis test was used for group comparisons of continuous variables and Fisher’s exact test for categorical variables. To identify the underlying homogeneous clusters of patients, we used a robust (nonparametric) unsupervised classification algorithm based on medoids (partitioning around medoids (PAM)) using the R package {cluster} [29]. Different from the widely used k-means clustering, rather than representing each cluster using a centroid (a vector of variable means, thus sensitive to outliers), each cluster is identified by its most representative observation (called a medoid). First, a random point is chosen and assigned as a medoid; the distance of every observation to each medoid is calculated, and each observation is assigned to its closest medoid; the “total cost” of the configuration is calculated as the sum of the distances obs-to-medoid; then, an iterative process picks a point that is not a medoid and reassigns it to be a medoid, with the total cost recalculated: if the cost is smaller, the new medoid is kept. The process is repeated until the medoids do not change. The included neuropsychological variables were standardized (Z scores) and centered before clustering because they were on different scales. The optimal number of clusters was determined by comparing the results of 26 different indices with the R package {NbClus} [30]. Variables included for clustering were IQ, coding, fluency, naming, visual and auditory attention, affect recognition, and theory of mind A and B. Multiple comparisons were corrected for false discovery rate using the Benjamini–Hockberg method [31]. R software v. 4.0.3 was used for analysis and graphics [32]. Statistical significance was set at *p* < 0.05.

## 3. Results

The demographic characteristics of the participants are reported in Table 1.

Cluster analysis revealed that three clusters explained 51.94% of point variability in performance in neuropsychological tasks (Figure 1).

We evaluated the clusters and determined that the three clusters could be described as follows: clusters 1 and 2, which respectively account for 35% and 48% of the sample, reflect normal global functioning, with cluster 1 showing more intra-individual variability and better performances only in IQ and executive functioning. The third group, which represented 17% of the sample, included children with poor performance in multiple domains.

The median scores of neuropsychological performances in the three groups are shown in Figure 2.

In the first two groups, all medical conditions were represented without significant differences between the groups. The first group (N = 40) included 15 healthy, 8 preterm, 6 HIE, and 11 CHD children. The second group (N = 55) included 14 healthy, 14 preterm, 18 HIE, and 9 CHD children.

On the contrary, the third group (N = 19) included only one healthy child, along with six preterm, four HIE, and eight CHD children. Univariate analysis and comparison of variables among clusters are reported in Table 2.

## 4. Discussion

In the present work, we investigate the possible emergence of distinct neuropsychological clusters in children with different adverse medical conditions (preterm, HIE, and CHD) and children without risk factors. Using an unsupervised, nonparametric, and very conservative cluster analysis, we found three emerging clusters that identified children with multiple impairments (cluster 3), children with neuropsychological profiles that were variable but in the average normal range (cluster 2), and children with adequate profiles and good performance in IQ and executive function tests (cluster 1). Interestingly, the clusters did not differ by medical status, with the only exception being that the pathological group comprised only one healthy child. For this study, healthy children were not recruited based on the absence of current neuropsychological dysfunctions but rather on the absence of pre/perinatal insults. Therefore, our healthy group reflects trends in the general population and the real occurrence of cognitive problems in children without known risk factors.

Our analysis grouped patients into different risk levels rather than identifying specific neuropsychological phenotypes, although an initial trend that highlighted differences in executive function and IQ quickly emerged. To our knowledge, this is the first study to evaluate outcomes after pre/perinatal insults from this point of view. Since different pathophysiological processes can lead to the same outcome, our unsupervised, data-driven approach tackles patients’ classification with a bottom-up approach. In this way, clear common ground is predefined before the analysis, which acts as an aid to describe the data-generating processes and not the contrary, as is usually done. This approach has previously been explored in significantly different contexts (i.e., aging) with very similar results. Rather than specific neuropsychological phenotypes, most studies identified relatively stable and high-functioning classes, contrasted by one or more clusters that declined to various extents (for a review, see [33]). Furthermore, the two classes (stable and high-functioning) represented the majority of the participants in most studies, as in our study.

The neurological processes of aging and early neurodevelopment are clearly quite different, as are the underlying neurobiological correlations. However, these studies begin to consider population heterogeneity. They identified latent cognitive trajectories by classifying and clustering participants with similar cognitive trajectories. This data-driven and hypothesis-free approach takes both intra-individual and inter-individual differences into account, which delineates the natural process of cognition more comprehensively.

In the neuroscience of human behavior and cognition, inter-individual differences are often treated as a source of ‘noise’ and, therefore, discarded by averaging data from a group of participants. However, inter-individual differences can be exploited to understand cognitive processes. Decades of research on sequela after common pre/perinatal insults have failed to identify specific phenotypes in the long-term sequelae of children experiencing stressful early conditions. Most of these studies did not consider inter-individual differences and compared results between subsamples classified according to clinical factors, hypothesizing that these factors contribute to differential trajectories. Another typical design used is to identify differences between two groups and commonalities between individuals. A similar design might have underestimated population heterogeneity. Although some areas of psychology, such as personality and intelligence research, have focused on inter-individual differences, this potentially powerful approach has rarely been used in the many years of cognition and neuropsychological studies in medical contexts. However, a large amount of information about the neural basis of human behavior and cognition can be obtained by specifically studying inter-individual variability [34].

Therefore, our data suggest that the high variability in results reported in the literature may not be a consequence of methodological issues or the paucity of work but may be a meaningful result in itself. It indicates different developmental trajectories characterized by good resilience to early stressful events with adequate development or by diffused vulnerability with a high risk of pervasive neurodevelopmental disorders, with the latter trajectory characterized by negative impacts on quality of life and, perhaps, academic achievement and the chance for a successful career and relational life in adulthood [35].

This understanding suggests that we can take advantage of the long developmental window of cerebral plasticity to mitigate or overcome the effects of perinatal brain insult effects. Cerebral modifications have been reported in children with reading problems and preterm children with learning and memory vulnerabilities [36,37], both showing increased microstructural organization after training. The ability to improve cerebral function reveals the considerable potential of behavioral assistance and rehabilitation programs to support remediation for children at risk of neurodevelopmental impairment. High-risk children may benefit from timely rehabilitative interventions that focus on the consolidation of developmental milestones that are essential for the subsequent maturation of more complex cognitive abilities. Timely intervention may have cumulative beneficial effects on cerebral circuitry. Obviously, a complete overview cannot avoid considering additional genetic and hereditary factors in the pathogenesis of neurodevelopmental disorders. How the genetic and epigenetic factors modulate the effect of risk and protective factors can help us develop personalized care and manage the conditions.

Two variables may have influenced our results. First, the choice of a nonhomogeneous sample for the medical status of the children. However, similar results have been found with a specific group [22]; in addition, patients from the three groups were similarly represented in the three clusters.

Second, our evaluation was conducted at a finite point in time and was not repeated. Using longitudinal cognitive data allowed the evaluation of ‘real neurodevelopmental trajectories’, highlighting the dynamic nature of cognition and its continuous change over time. Furthermore, repeated evaluations that would have allowed a better investigation of executive functions were not possible in our investigation because the children were too young to explore these functions extensively. Until eight years of age, many executive functions are still immature (e.g., flexibility and inhibition), and few children perform executive tasks reliably before that age. Given this limitation, specific trends of executive functions may have been masked in our study. Although there is heterogeneity in the results, most works seem to converge to identify vulnerabilities related to these complex high-order functions. If difficulty with executive function is a common cognitive complaint, it may be a key early symptom of neurodevelopmental vulnerability, and strong executive function may be the strength that allows one group of children to develop global resilience (perhaps our children in cluster 1). Therefore, the fact that our study did not thoroughly investigate executive function may mean that it underrecognized a slight trend of difference that emerged between the two main clusters. This possibility is highlighted by the differences found between clusters 1 and 2, which might evolve into a more distinct executive profile over time.

An important limitation of our study is that we did not consider risk factors originating in the earlier pregnancy, such as maternal smoking and alcohol consumption, maternal diabetes, maternal drug dependency, and malnutrition. Furthermore, the crucial aspects of child–mother interactions and parental competencies were not sufficiently explored because we only excluded the presence of major problems (trauma, parental neglect, and parental pathologies). These conditions are well known to contribute to neurodevelopmental problems.

It could be interesting to investigate how these risk factors could influence neurodevelopment and how protective factors (e.g., healthy lifestyle, social support, etc.) could mitigate the negative effects of pre/perinatal injury on development.

Finally, two limitations of our study are the number of patients enrolled and the algorithm performance: a potential limitation of our approach is, in fact, the confined provenience of our patients/data, which could have overestimated the performance of our clustering methods. Larger and wider studies are needed to refine the clustering algorithm and check its real-world performance. We hope that future research will explore neurodevelopmental trajectories by recruiting a higher number of participants and by considering other influencing variables.

## 5. Conclusions

In conclusion, the causative mechanisms for adverse neuropsychological outcomes are multifactorial, interrelated, cumulative, and likely synergistic over time. Brain maturation, including the refinement of brain networks and myelination, continues through childhood, providing a significant window for highly variable outcomes but also for recovery. As a result, an early assessment of risk indicators is critical. Unfortunately, the use of behavioral tasks to identify risk indicators poses problems of specificity (i.e., reducing the rate of false positives) and sensitivity (i.e., reducing the rate of false negatives) for these instruments. Advancements in research may come from studying the neurobiological mechanisms underlying cognition. The identification of neurobiological measures implicated in the development of neurological functions can provide considerable advancement in the research on early risk indicators, with promises related to the mitigation of pre/perinatal insults [6,15,38,39].

## Figures and Tables

**Figure 1 children-09-00775-f001:**
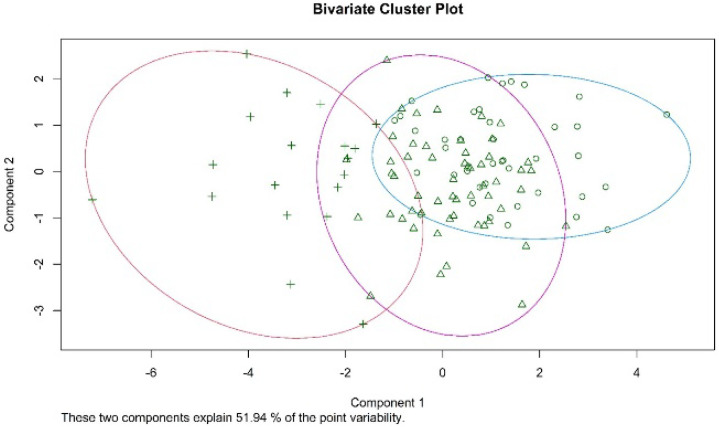
The three clusters explaining 51.94% of the point variability of performance in neuropsychological tasks. Different clusters belonging is depicted as different shapes of the observations (plus sign, triangle, and circle).

**Figure 2 children-09-00775-f002:**
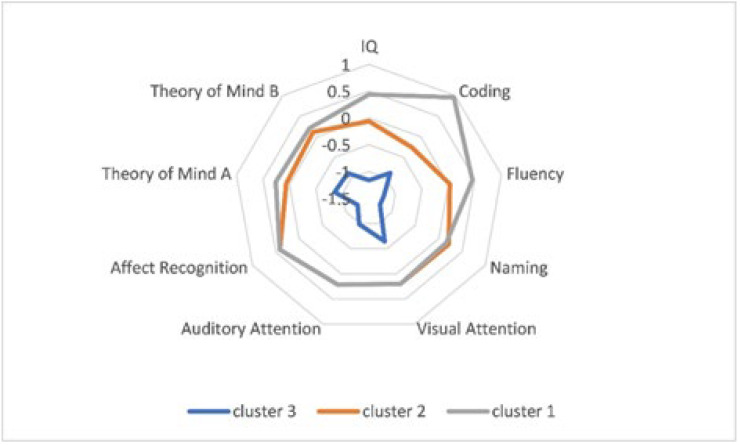
Median scores of neuropsychological performances in the three clusters.

**Table 1 children-09-00775-t001:** Demographic characteristics of the participants. Age is expressed in months (IQR).

Group	Number	Males	Age
Healthy	30	18 (60%)	112.50 (78.25, 127.00)
Preterm	28	17 (60, 70%)	73.00 (70.25, 91.00)
HIE	28	18 (64, 30%)	71.00 (65.00, 83.25)
CHD	28	14 (50%)	92.50 (69.00, 96.00)

Legend: HIE: hypoxic-ischemic encephalopathy; CHD: congenital heart disease.

**Table 2 children-09-00775-t002:** Univariate analysis and variables comparisons among clusters.

	Cluster			Adj. *p*-Values		
Characteristic	1, N = 40 ^1^	2, N = 55 ^1^	3, N = 19 ^1^	1 vs. 2 ^2^	1 vs. 3 ^2^	2 vs. 3 ^2^
IQ	0.44 (−0.06, 1.02)	−0.06 (−0.50, 0.40)	−1.00 (−1.60, −0.60)	0.022	<0.001	<0.001
Coding	0.96 (0.35, 1.57)	−0.26 (−0.86, 0.05)	−0.56 (−1.47, −0.26)	<0.001	<0.001	0.045
Fluency	0.46 (−0.06, 1.14)	0.03 (−0.44, 0.61)	−1.17 (−1.54, −0.55)	0.2	<0.001	<0.001
Naming	0.16 (−0.23, 0.61)	0.21 (−0.22, 0.68)	−1.41 (−1.66, −0.85)	0.7	<0.001	<0.001
Visual Attention	0.20 (−0.16, 0.57)	0.20 (−0.35, 0.57)	−0.16 (−1.26, 0.02)	0.3	0.002	0.045
Auditory Attention	0.21 (0.21, 1.00)	0.21 (−0.57, 1.00)	−0.57 (−1.74, −0.57)	0.3	<0.001	<0.001
Affect Recognition	0.41 (−0.12, 0.68)	0.41 (−0.25, 0.68)	−1.19 (−1.99, −0.79)	0.9	<0.001	<0.001
Theory of Mind A	0.26 (−0.07, 0.87)	0.06 (−0.45, 0.69)	−0.85 (−1.55, −0.22)	0.6	<0.001	0.001
Theory of Mind B	0.21 (−0.30, 0.93)	0.12 (−0.43, 0.63)	−0.81 (−1.23, −0.13)	0.7	<0.001	0.001
Age (months)	81 (71, 106)	75 (68, 110)	93 (73, 96)	0.7	0.8	0.5
Group				0.3	0.063	0.066
CHD	11 (28%)	9 (16%)	8 (42%)			
Controls	15 (38%)	14 (25%)	1 (5.3%)			
HIE	6 (15%)	18 (33%)	4 (21%)			
Preterms	8 (20%)	14 (25%)	6 (32%)			
Sex	17 (42%)	26 (47%)	6 (32%)	0.7	0.5	0.3

^1^ Median (IQR); n (%). ^2^ Benjamini–Hochberg correction for multiple testing. Legend: CHD: congenital heart disease; HIE: hypoxic-ischemic encephalopathy.

## Data Availability

Data supporting the findings of this study are available from the corresponding author upon reasonable request.

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
