# Peer review of "Embrace the Complexity: Agnostic Evaluation of Children’s Neuropsychological Performances Reveals Hidden Neurodevelopment Patterns"

_children, 2022, doi:10.3390/children9060775_

Round 1

Reviewer 1 Report

The authors performed detailed and age-appropriate cognitive and neuropsychological investigations in 114 children aged 5-10 years old. The sample was selected from earlier studies performed by the authors in children treated as neonates or young children on an intensive care unit, either due to prematurity, neonatal asphyxia or surgery for a cardiac defect. Former healthy newborns served as a control group. Children with definitive and severe pathology such as severe mental retardation, cerebral palsy, epilepsy, brain malformations or genetic syndromes were excluded from the analysis. Applying Cluster analysis to the neuropsychological data the authors could identify 3 distinct clusters of outcome: clusters 1 and 2, reflected normal global functioning, with cluster 1 showing more intra-individual variability and better performances only in IQ and executive functioning. The third group, which represented 17% of the sample, included children with poor performance in multiple domains. Between cluster 1 and 2 the distribution of risks groups and normal children did not differ, whereas in cluster 3 the former healthy newborns were significantly underrepresented. The authors conclude that no specific pattern of neuropsychological deficits can be attributed to the different risk groups, but that the residual deficits are very individual pointing to additional individual risk and resilience patterns in their pathogenesis.

The manuscript presents an interesting new statistical approach to such material and research question and in my opinion, it should be published. It confirms results that have been found many years ago in large field studies, for example from Esser and Schmidt in Germany. However, I have some points to discuss that should be considered to improve the manuscript.

  1. I recommend to describe the clinical findings of the 3 risk groups in detail; it is not sufficient to refer to the earlier publications because not each reader will have access to these publications. How many of the prematures were very-low-birth-weight or > 2.00 g bw, respectively; how many of them suffered amniotic infection syndrome which is regarded as major reason for neurological and cognitive sequelae; How many of the asphyxiated infants were severely asphyxiated and showed post-asphyctic encephalopathy? How many of the cardiac children suffered a cyanotic vitium and how many were operated in extracorporeal circulation or with hypothermia? Noteworthy, the cardiac defect children were not all operated in the perinatal period but up to an age of 5 years.
  2. The case numbers in the older publications are not identical to the present ones. Please declare this.
  3. What does the term pre-perinatal injury mean. Injury arising before the perinatal period or before and in the perinatal period? Noteworthy, there is no risk group included in this study which originates in the earlier pregnancy, although conditions such as maternal smoking and alcohol consumption, maternal diabetes, maternal drug dependency and malnutrition are well know to contribute to neurodevelopmental problems in the off-spring. I advise to at least include this aspect in the discussion; but it would be even better if these risk factors were known and could be included in the analysis itself.
  4. Please include discussion of familial and hereditary factors in the pathogenesis of neuropsychological dysfunction in children. There is a lot of evidence that dysfunctions such as ADS, autism and legasthenia have very strong hereditary backgrounds and a lot of involved genes are already known.
  5. What about psycho-social risk factors in pregnancy and early infancy? Mother-child interaction? This apparently also plays a major role in early neurodevelopment.
  6. I personally do not believe that the frequency of neurodevelopmental problems has really increased so much as the authors write in their introduction: I think that these children have always been there, but that awareness of the problem, stressful conditions at school and in general life, and diagnostic possibilities have increased contributing a lot to the increasing incidence rates.
  7. I do not believe that these neurodevelopmental problems are largely overlooked in daily practice. Since decades pediatricians have developed measures for early diagnosis and in many countries neurodevelopmental centers are active for these children.
  8. Last but not least: There is no question that these children with neurodevelopmental problems have to be cared for. However, I am not so optimistic as the authors to believe that very early treatment can improve their functioning a lot, it is more a question of acceptance and adequate handling of the children. There are several Cochrane reviews showing that by very early NICU- or post-NICU stimulation and enrichment the interaction and cognitive function of infants can be improved forthe first years of age, but that at school age these positive effects are lost.

Author Response

Reviewer 1

The authors performed detailed and age-appropriate cognitive and neuropsychological investigations in 114 children aged 5-10 years old. The sample was selected from earlier studies performed by the authors in children treated as neonates or young children on an intensive care unit, either due to prematurity, neonatal asphyxia or surgery for a cardiac defect. Former healthy newborns served as a control group. Children with definitive and severe pathology such as severe mental retardation, cerebral palsy, epilepsy, brain malformations or genetic syndromes were excluded from the analysis. Applying Cluster analysis to the neuropsychological data the authors could identify 3 distinct clusters of outcome: clusters 1 and 2, reflected normal global functioning, with cluster 1 showing more intra-individual variability and better performances only in IQ and executive functioning. The third group, which represented 17% of the sample, included children with poor performance in multiple domains. Between cluster 1 and 2 the distribution of risks groups and normal children did not differ, whereas in cluster 3 the former healthy newborns were significantly underrepresented. The authors conclude that no specific pattern of neuropsychological deficits can be attributed to the different risk groups, but that the residual deficits are very individual pointing to additional individual risk and resilience patterns in their pathogenesis.

The manuscript presents an interesting new statistical approach to such material and research question and in my opinion, it should be published. It confirms results that have been found many years ago in large field studies, for example from Esser and Schmidt in Germany. However, I have some points to discuss that should be considered to improve the manuscript.

  1. I recommend to describe the clinical findings of the 3 risk groups in detail; it is not sufficient to refer to the earlier publications because not each reader will have access to these publications. How many of the prematures were very-low-birth-weight or > 2.00 g bw, respectively; how many of them suffered amniotic infection syndrome which is regarded as major reason for neurological and cognitive sequelae; How many of the asphyxiated infants were severely asphyxiated and showed post-asphyctic encephalopathy? How many of the cardiac children suffered a cyanotic vitium and how many were operated in extracorporeal circulation or with hypothermia? Noteworthy, the cardiac defect children were not all operated in the perinatal period but up to an age of 5 years.

We thank the reviewer for this comment. We created appendix 1, with all the relevant clinical data and inclusion and exclusion criteria for the study.

  1. The case numbers in the older publications are not identical to the present ones. Please declare this.

Done.

  1. What does the term pre-perinatal injury mean. Injury arising before the perinatal period or before and in the perinatal period? Noteworthy, there is no risk group included in this study which originates in the earlier pregnancy, although conditions such as maternal smoking and alcohol consumption, maternal diabetes, maternal drug dependency and malnutrition are well know to contribute to neurodevelopmental problems in the off-spring. I advise to at least include this aspect in the discussion; but it would be even better if these risk factors were known and could be included in the analysis itself.

We used the term pre-perinatal injury because the study includes prenatal conditions (prematurity) and perinatal conditions (HIE); or, like CHD, conditions that could affect both the fetus's well-being and complicate labor.

Unfortunately, our study doesn’t comprise risk factors associated with the mother's lifestyle or conditions. This is an important limitation of our work, and we will consider this important aspect for future research. We added this comment to the discussion:

“An important limitation of our study is that we didn’t consider risk factors originat-ing in the earlier pregnancy, such as maternal smoking and alcohol consumption, maternal diabetes, maternal drug dependency, and malnutrition. Furthermore, the crucial aspects of child-mother interactions and parental competencies have not been sufficiently explored because we only excluded the presence of major problems (trauma, parental neglect, parental pathologies). These conditions are well known to contribute to neurodevelopmental problems.

 It could be interesting to investigate how these risk factors could influence neurodevelopment and how protective factors (e.g., healthy lifestyle, social support, etc.) could mitigate the negative effect of pre-perinatal injury on development.”

  1. Please include discussion of familial and hereditary factors in the pathogenesis of neuropsychological dysfunction in children. There is a lot of evidence that dysfunctions such as ADS, autism and legasthenia have very strong hereditary backgrounds and a lot of involved genes are already known.

We thank the reviewer for the comment. The following part has been added to the discussion:

“Obviously, a complete overview cannot avoid considering also genetic and hereditary factors in the pathogenesis of neurodevelopmental disorders. How the genetic and epigenetic factors modulate the effect of risk and protective factors can help develop personalized care and manage the conditions.”

  1. What about psycho-social risk factors in pregnancy and early infancy? Mother-child interaction? This apparently also plays a major role in early neurodevelopment.

We thank the reviewer for the important comment. Unfortunately, we have only superficially considered this aspect. We administered the PSI-SF (Parenting stress questionnaire) and we conducted a structured interview with the parents to evaluate the presence of neglect, parental pathologies, or other traumatic events. However, we did not evaluate the mother-child interaction or parental competencies with specific instruments; this has been added as a limit in the discussion:

“Furthermore, the crucial aspects of child-mother interactions and parental competencies have not been sufficiently explored because we only excluded the presence of major problems (trauma, parental neglect, parental pathologies). These conditions are well known to contribute to neurodevelopmental problems.”

  1. I personally do not believe that the frequency of neurodevelopmental problems has really increased so much as the authors write in their introduction: I think that these children have always been there, but that awareness of the problem, stressful conditions at school and in general life, and diagnostic possibilities have increased contributing a lot to the increasing incidence rates.

This is part of a debate with a very long history, particularly for some disorders like autism. In parallel, in several fields of medicine not directly associated with neuropsychiatric disorders, like cardiology, the scientific community's attention has been moved from survival to the quality of life of the patient, given the great advance in medical techniques. So, whether the increase is real or is the reflex of major attention to it, a better understanding of neurodevelopmental trajectories becomes central. 

  1. I do not believe that these neurodevelopmental problems are largely overlooked in daily practice. Since decades pediatricians have developed measures for early diagnosis and in many countries neurodevelopmental centers are active for these children.

We agree with the reviewer; in the last years, a great effort has been made both in diagnosing and managing several neuropsychiatric conditions. But this is mostly true for conditions like autism and genetic syndromes. Instead, we have observed how often the neurocognitive impairments associated with some medical conditions are considered an ancillary report. For example, among neurological conditions, epilepsy is one in which cognition is most recognized: actually, epilepsy and cognition are considered two symptoms of the same neurobiological vulnerability. Although that, the Italian health system doesn’t guarantee a cognitive evaluation for all epileptic patients, and only advanced centers consider these aspects. So is for congenital heart disease and perinatal asphyxia.

  1. Last but not least: There is no question that these children with neurodevelopmental problems have to be cared for. However, I am not so optimistic as the authors to believe that very early treatment can improve their functioning a lot, it is more a question of acceptance and adequate handling of the children. There are several Cochrane reviews showing that by very early NICU- or post-NICU stimulation and enrichment the interaction and cognitive function of infants can be improved forthe first years of age, but that at school age these positive effects are lost.

Thanks for the comment. We, in part, agree with the reviewer, but, in part, we want to believe that something can be done. We also believe that a great role has the entity of the insult and the presence of structural or functional abnormalities.

Reviewer 2 Report

Dr. Elisa Cainelli and his colleagues performed an interesting unsupervised cluster analysis study related to the impact of early perinatal adverse events on 114 children aged 5-10 years healthy or with perinatal diagnoses such as prematures, congenital heart diseases, and hypoxic ischemic encephalopathy. General intelligence, language, attention memory, executive functions, and social skills have been evaluated with the emergence of 3 clusters based on severity levels: 1. multiple impairments, 2. variable neuropsychological profiles within the normal range,  and 3.adequate/good IQ performance

This study is very  interesting, however, I have some concerns that hinder the publication in the current form

#1 Abstract

Minor

The authors need to specify hypoxic-ischemic encephalopathy (HIE) in addition to CHD, which has already been mentioned. It is not clear from the beginning what entities from the neonatal encephalopathy syndrome have been studied. In fact, the reader has to infer from the keywords that they have studied the impact of HIE, which is mentioned later on in the introduction.

#2 Introduction and discussions

Minor

  1. Line 70

The authors should succinctly present the pro-inflammatory processes they refer to. For example, you could use Neamtu et al paper  A Decision-Tree Approach to Assist in Forecasting the Outcomes of the Neonatal Brain Injury Int. J. Environ. Res. Public Health 2021, 18(9), 4807; https://doi.org/10.3390/ijerph18094807, or any other manuscript in this respect:

„ At a molecular level, many pathways were incriminated in apoptosis of premyelinating oligodendrocytes or subplate neurons involved in perinatal brain development. Glutamate rising concentrations or free radical reactive species (both oxygen and hydrogen) in HIE, inflammatory cytokines such as TNF-α, IL-1b, IL-6, 12, 15, 18 from activated microglia and astrocytes, and low pH in INF, free iron secondary to HC were extensively mentioned in both white and grey matter injuries. „

Different pathophysiological processes lead to the same outcome eventually, namely white and grey matter, so it is crucial to be classified by severity levels as a starting point in further evaluations. The authors’ approach based on an unsupervised machine learning design adds value to the literature focusing on the supervised machine learning approaches (see the aforementioned report). This could also be specified  in the discussion section when discussing the machine learning approaches

  1. Line 83

Given the lack of severe impairment...the authors probably wanted to say given the mild or moderate levels of impairment  ...these children were often considered normal healthy or in general without problems. Therefore they have not been followed and monitored on a long-term neurodevelopmental basis.

#3 Materials and methods

Major

The inclusion criteria should be succinctly presented in an appendix since they have 3 different clusters, and each of mentioned papers in the references(6,14, 21) refer to a different group. It would be more elegant and preferable for the reader to read a single section that sums up these inclusion criteria than to look for different papers and to try to figure out the big picture

Minor

Lines 140-142 present the results and should be mentioned in the results section. Moreover, the table legend should specify what it is presented regarding the age (mean/median, minimum, maximum)

Minor

2.3 Statistical Analysis

Minor

 the α level should be specified

Major

The algorithm mechanism should be presented in brief with a focus on the general audience's comprehension (please notice the reference 21)

Did the authors test the accuracy of the unsupervised machine learning algorithm? How did they assess it? Rand index, perhaps? If they didn’t then, they should mention it as a study limitation and/or future perspectives

#4 Discussion

Major

Limitations regarding the number of enrolled subjects and the algorithm performance should be addressed. Future research perspectives should be presented more clearly in this part rather than a phrase in conclusion.

Author Response

Reviewer 2

Dr. Elisa Cainelli and his colleagues performed an interesting unsupervised cluster analysis study related to the impact of early perinatal adverse events on 114 children aged 5-10 years healthy or with perinatal diagnoses such as prematures, congenital heart diseases, and hypoxic ischemic encephalopathy. General intelligence, language, attention memory, executive functions, and social skills have been evaluated with the emergence of 3 clusters based on severity levels: 1. multiple impairments, 2. variable neuropsychological profiles within the normal range,  and 3.adequate/good IQ performance

This study is very  interesting, however, I have some concerns that hinder the publication in the current form

#1 Abstract

Minor

The authors need to specify hypoxic-ischemic encephalopathy (HIE) in addition to CHD, which has already been mentioned. It is not clear from the beginning what entities from the neonatal encephalopathy syndrome have been studied. In fact, the reader has to infer from the keywords that they have studied the impact of HIE, which is mentioned later on in the introduction.

We thank the reviewer for the comment. We were unaware of the confusing use of HIE and perinatal asphyxia we made. The term HIE is now consistently used throughout the manuscript, and perinatal asphyxia has been removed.

#2 Introduction and discussions

Minor

  1. Line 70

The authors should succinctly present the pro-inflammatory processes they refer to. For example, you could use Neamtu et al paper  A Decision-Tree Approach to Assist in Forecasting the Outcomes of the Neonatal Brain Injury Int. J. Environ. Res. Public Health 2021, 18(9), 4807; https://doi.org/10.3390/ijerph18094807, or any other manuscript in this respect:

„ At a molecular level, many pathways were incriminated in apoptosis of premyelinating oligodendrocytes or subplate neurons involved in perinatal brain development. Glutamate rising concentrations or free radical reactive species (both oxygen and hydrogen) in HIE, inflammatory cytokines such as TNF-α, IL-1b, IL-6, 12, 15, 18 from activated microglia and astrocytes, and low pH in INF, free iron secondary to HC were extensively mentioned in both white and grey matter injuries. „

We thank the reviewer for the comment. We added this part to the introduction as suggested:

“Infection or inflammation can sensitize the brain to injuries and make them more severe. Cytokines such as IL-10 are inflammatory mediators that readily cross the blood-brain barrier. They have a dual role of being protective and harmful, and their balance is critical in brain injury, even if their increments are non-specific. Among cytokines, IL-1beta promotes brain damage through reactive oxygen species rise and, with IL-6 and TNF-alpha, its level is a proxy of white matter injury that could lead to cerebral palsy when measured in the amniotic fluid of women with complicated preterm gestation. Many pathways were incriminated in apoptosis of premyelinating oligodendrocytes or subplate neurons involved in perinatal brain development at a molecular level. Glutamate rising concentrations or free radical reactive species (both oxygen and hydrogen) in HIE, and inflammatory cytokines from activated microglia and astrocytes were extensively mentioned in both white and grey matter injuries [9,10].”

Different pathophysiological processes lead to the same outcome eventually, namely white and grey matter, so it is crucial to be classified by severity levels as a starting point in further evaluations. The authors’ approach based on an unsupervised machine learning design adds value to the literature focusing on the supervised machine learning approaches (see the aforementioned report). This could also be specified  in the discussion section when discussing the machine learning approaches

The following sentences have been added to the discussion:

Since different pathophysiological processes could lead to the same outcome, our unsupervised, data-driven approach tackles patients’ classification with a bottom-up approach. In this way, a clear common ground is pre-defined before the analysis that acts as an aid to describe the data-generating processes and not the contrary as usually done.”

  1. Line 83

Given the lack of severe impairment...the authors probably wanted to say given the mild or moderate levels of impairment  ...these children were often considered normal healthy or in general without problems. Therefore they have not been followed and monitored on a long-term neurodevelopmental basis.

We rewrote the sentences as follows:

“Given the lack of severe impairment, the lack of confidence of the medical professional with neurocognitive problems, and the difficulty in managing such long-distance emerg-ing symptoms, these children were often considered "normal", "healthy," or in general without problems. Therefore, they have not been followed and monitored on a long-term neurodevelopmental basis.”

#3 Materials and methods

Major

The inclusion criteria should be succinctly presented in an appendix since they have 3 different clusters, and each of mentioned papers in the references(6,14, 21) refer to a different group. It would be more elegant and preferable for the reader to read a single section that sums up these inclusion criteria than to look for different papers and to try to figure out the big picture

We agree with the reviewer; we added the inclusion and exclusion criteria, together with the clinical data of the samples, in an appendix (appendix 1).

Minor

Lines 140-142 present the results and should be mentioned in the results section. Moreover, the table legend should specify what it is presented regarding the age (mean/median, minimum, maximum)

We thank the reviewer for the comment. The table has been moved to the results section, and information about age was added.

Minor

2.3 Statistical Analysis

Minor

 the α level should be specified

Done.

Major

The algorithm mechanism should be presented in brief with a focus on the general audience's comprehension (please notice the reference 21)

The following sentences have been added in the statistical analysis section:

“ Different from the largely used k-means clustering, rather than representing each cluster using a centroid (a vector of variable means thus sensitive to outliers), each cluster is identified by its most representative observation (called a medoid). First, a random point is chosen and assigned as a medoid; the distance of every observation to each medoid is calculated, and each observation is assigned to its closest medoid; the “total cost” of the configuration is calculated as the sum of the distances obs-to-medoid; here starts an iterative process that picks a point that is not a medoid and reassigns it to be a medoid, with the total cost recalculated: if the cost is smaller the new medoid is kept. The process is repeated until the medoids don’t change. “

Did the authors test the accuracy of the unsupervised machine learning algorithm? How did they assess it? Rand index, perhaps? If they didn’t then, they should mention it as a study limitation and/or future perspectives

We thank the reviewer for the question, but since we had no “true” cluster to which assess accuracy, we could have only reported the percentage of the variability described by our clustering method (already in the manuscript). We added the phrase:

“A potential limitation of our approach is the confined provenience of our patients/data that could have overestimated the performance of our clustering methods. Larger and wider studies are needed to refine the clustering algorithm and check its real-world performances.”

#4 Discussion

Major

Limitations regarding the number of enrolled subjects and the algorithm performance should be addressed. Future research perspectives should be presented more clearly in this part rather than a phrase in conclusion.

We agree with the reviewer. We added this part to the discussion:

“Finally, two limitations of our study are the number of patients enrolled and the algorithm performance: a potential limitation of our approach is, in fact, the confined provenience of our patients/data, which could have overestimated the performance of our clustering methods. Larger and wider studies are needed to refine the clustering algorithm and check its real-world performances. We hope that future research will explore neuro-developmental trajectories by recruiting a higher number of participants and by considering other influencing variables.”

Round 2

Reviewer 1 Report

The authors have addressed all my comments adequately.

Reviewer 2 Report

The authors successfully addressed every issue raised in the previous report. Their approach is very interesting and adds value to the current literature. The manuscript clarity improved significantly, therefore I recommend its publication.